# Social support and psychosocial well-being among older adults in Europe during the COVID-19 pandemic: a cross-sectional study

Ji Lu [ORCID],[1] Juyang Xiong [ORCID],[1] Shangfeng Tang [ORCID],[1] Ghose Bishwajit [ORCID],[2] Shuyan Guo[3]

JL and JX contributed equally.

[1]School of medicine and health management, Tongji Medical College, Huazhong university of science and technology, Wuhan, China
[2]Faculty of health sciences, University of ottawa, Ottawa, Ontario, Canada
[3]National Institute of Hospital Administration, National Health Commission, Beijing, China

**Correspondence to**
Shuyan Guo;
guoshuyan@niha.org.cn

## ABSTRACT

The objective of the study was to identify the association between social support and psychosocial well-being among men and women aged over 65 years in Europe during the COVID-19 pandemic.

**Methods** Cross-sectional data on 36 621 men (n=15 719) and women (n=20,902) aged 65 years or higher were obtained from the ninth round of the Survey of Health, Ageing and Retirement in Europe. The outcomes were measured by psychosocial well-being reflected with self-reported depression, nervousness, loneliness and sleep disturbances. Social support was measured in terms of receiving help from own children, relatives and neighbours/friends/colleagues since the pandemic outbreak.

**Result** About one-third of the participants reported depression (31.03%), nervousness (32.85%), loneliness (32.23%) and sleep trouble (33.01%). The results of multivariable regression analysis revealed that social support was a protective factor to psychological well-being. For instance, receiving help from own children (RD=−0.13, 95% CI=−0.14 to −0.12), relatives (RD=−0.08, 95% CI=−0.11 to −0.06), neighbours/friends/colleagues (RD=−0.11, 95% CI=−0.13 to −0.09) and receiving home care (RD=−0.20, 95% CI=−0.22 to −0.18) showed significantly lower risk difference for depression. Similar findings were noted for loneliness, nervousness, and sleep trouble as well, with the risk difference being slightly different for men and women in the gender-stratified analysis. For instance, the risk difference in depression for receiving help from own children was −0.10 (95% CI=−0.12 to −0.08) among men compared with −0.12 (95% CI=−0.14 to −0.11) among women. The risk differences in the outcome measures were calculated using generalised linear model for binomial family.

**Conclusion** Findings of the present study highlight a protective role of social support on psychological well-being among both men and women. Developing strategies to promote social support, especially among older adults, may mitigate the rising burden of psychological illness during the COVID-19 pandemic.

## STRENGTHS AND LIMITATIONS OF THIS STUDY

⇒ This study focused on the association between social support and psychosocial well-being among older adults during the COVID-19 pandemic in European Union countries.
⇒ The analysis conducted using multiple dimensions of psychosocial well-being and social support.
⇒ The cross-sectional nature of the surveys prevents establishing causal relationships between outcome and explanatory variables.
⇒ Inclusion of all relevant sociocultural factors was not possible due to the secondary analysis.

## INTRODUCTION

The COVID-19 pandemic has become one of the most devastating health and social crises in generations due to its long-term impacts on physical and mental health, social life and livelihoods worldwide.[1–3] A pervasive crisis is the emergence of a phenomenon known as 'coronaphobia'—fear or anxiety related to the virus and its potential impacts on the psychosocial well-being of individuals and communities at large.[4 5] A growing number of studies have highlighted the prevalence of mental illness in the context of lifestyle such as higher consumption of psychoactive drugs and widespread social isolation and loneliness.[6–9] The fear of contracting the virus has compelled people to stay at home for an extended period of time, particularly affecting older adults who face an elevated risk of severe illness from the virus.[10]

The measures to control the COVID-19 pandemic, such as physical distancing, have caused disruptions to a healthy and active lifestyle, as well as the opportunities to maintain social connections and seek help in times of need.[11] The impact of social isolation and loneliness on mental health has been an increasing concern in the wake of the global pandemic, as socialisation opportunities have become limited and physical distancing rules have been put in place.[12] As such, there has been a dramatic decrease in social interaction

and support, which is particularly concerning for peoples' psychosocial well-being during the global health crisis. Prolonged isolation can lead to feelings of loneliness, depression and anxiety and can take a toll on mental health of older adults due to their higher dependence on physical and emotional care.[13 14]

The feeling of depression and loneliness are common mental health concerns that can affect people from all walks of life, especially among older adults. This is due to a range of factors, including changes in physical health and lifestyle, increased isolation and loneliness, financial concerns and difficulty in adapting to the technological advances of modern life. The COVID-19 pandemic has fuelled social isolation and loneliness mainly through fewer opportunities for meeting friends and families, organising or participating in social gatherings and reduced social support.[15] Social support, which is a key aspect of social capital, has been shown to be strongly correlated with mental health in previous studies.[16 17] Perceived social support refers to the subjective availability of psychological, physical and financial help from social relationships such as family, friends, neighbours and community members that is available in times of need. Social support plays an important role in mitigating the impact of social isolation and can help reduce stress, improve mood, and promote positive coping mechanisms.[18] Social support also provides a sense of belonging and connection, which can offset feelings of loneliness and isolation. Several previous studies have demonstrated the link between social relationships and support and mental health.[16 18 19] However, there is not enough evidence regarding this relationship among older adults in European countries in the context of the COVID-19 pandemic. The present study, therefore, investigates the relationship between social support and psychosocial well-being among older adults in Europe using the data from the Survey of Health Ageing, and Retirement (wave 9).

## METHODS
### Data source
Data for this study were derived from the ninth wave of the Survey of Health Ageing, and Retirement in Europe (SHARE).[20] The survey aims to provide a framework for researchers to better understand population ageing and covers most of the European Union (EU) and Israel. SHARE has been adopted as a model by a number of ageing surveys throughout the world and is harmonised with the US Health and Retirement Study and the English Longitudinal Study of Ageing.[21] The SHARE survey collects data on a diverse range of health and socioeconomic indicators among individuals including demographics, living conditions, income and employment status, mental health, satisfaction with health conditions and social networks. The ninth wave of the survey included questions relevant to COVID-19, such as receiving support from social connections to obtain

basic necessities. Data for this survey were collected via computer-assisted telephone interviews between June and August 2021 among individuals aged 50 or older. For the purpose of the present study, the analysis was limited to participants aged 65 years or older only.

### Description of the variables
The outcome measures included four indicators of psychosocial well-being assessed by self-reported nervousness, depression, sleep disturbances and loneliness. These indicators were assessed by the following questions: (1) felt nervous in last month (yes/no), (2) sad or depressed in last month (yes/no), (3) trouble sleeping recently (yes/no) and (4) how often feelings of loneliness (often/some of the times/hardly ever or never). For loneliness, the answers were recoded to 'yes' for responses 'often/some of the times' and to 'no' for responses 'Hardly ever or never'. Social support was assessed in terms of receiving help in obtaining necessities during the pandemic from own children, relatives and neighbours/friends/colleagues to obtain necessities since the outbreak. These indicators were assessed by the following questions: (1) help received from own children to obtain necessities since the outbreak (yes/no), (2) help received from other relatives to obtain necessities since the outbreak (yes/no), (3) help received from neighbours/friends/colleagues to obtain necessities (yes/no) and (4) received regular home care during last 3 months (yes/no). Apart from the indicators of social support, the analysis also included several socioeconomic variables based on their known association with mental health status such as age (65–69 years/70–74 years/75–79 years/80–84 years/85+ years), sex (male/female), employment situation (retired/employed or self-employed/unemployed/permanently sick or disabled? homemaker/other), and country of residence (list of countries provided in online supplemental appendix).

## STATISTICAL ANALYSIS
All statistical analyses were conducted using Stata V.17 (StataCorp). The dataset was first examined to make sure the sample population was defined correctly (65 years and older) and there were no missing values and outliers. Descriptive statistics were carried out to calculate the percentage of participants reporting nervousness, depression, loneliness and sleep trouble. Sex and country-level percentages were presented as bar charts. In the next step, we conducted generalised linear models (GLMs) for binomial family to measure the risk difference (RD) (with 95% CIs) in the four outcome variables (nervousness, depression, loneliness and sleep trouble). The results of GLMs were presented as RD, defined as the risk of a condition between an exposed group and an unexposed group.[22] A positive RD value indicates a higher risk by the exposure, whereas a negative one indicates a lower risk. Given the well-documented gender differences in mental health outcomes, the analysis was stratified by male and

**Table 1** Percentage of participants who reported having depression, nervousness, loneliness and sleep trouble by sex

|  | Percentage of total | Percentage of total men | Percentage of total women |
|---|---|---|---|
| Depression | 31.03 | 22.92 | 37.12 |
| Nervousness | 32.85 | 26.44 | 37.67 |
| Loneliness | 32.23 | 23.57 | 38.73 |
| Sleep trouble | 33.01 | 25.73 | 38.49 |

female participants for each outcome variable. The level of statistical significance for all associations was set at 0.05.

## PATIENT AND PUBLIC INVOLVEMENT
No patient involved.

## RESULTS
The sociodemographic characteristics of the sample population are available in the online supplemental appendix file. A greater percentage of the participants were in the age bracket of 65–69 years (n=26.92%), female (n=57.08%), retired (n=88.03%) and rated general health status as good (n=39.05%). Sample size for the participating countries was as follows: Austria (n=1943), Germany (n=1466), Sweden (n=856), Netherlands (n=592), Spain (n=1545), Italy (n=2528), France (n=1458), Denmark (n=1143), Greece (n=2515), Switzerland (n=1452), Belgium (n=2427), Israel (n=1132), Czech Republic (n=1808), Poland (n=1795), Luxembourg (n=581), Hungary (n=715), Portugal (n=860), Slovenia (n=2313), Estonia (n=3012), Croatia (n=1305), Lithuania (n=774), Bulgaria (n=480), Cyprus (n=490), Finland (n=855), Latvia (n=623), Malta (n=548), Romania (n=919) and Slovakia (n=486).

Among the four psychosocial well-being indicators, sleep trouble (33.01%) was the most commonly reported one (table 1), followed by nervousness (32.85%) and loneliness (32.23%). As described in table 1, the percentage of reporting depression (77.08% vs 62.88%), nervousness (73.56% vs 62.33%) and loneliness (76.43% vs 61.27%) was higher in men, while that of sleep trouble was higher in women (25.73% vs 38.49%).

The results of multivariable regression analysis on the association between self-reported depression, nervousness, loneliness, sleep trouble and social support are shown in table 2. Receiving help from own children (RD=−0.13, 95% CI=−0.14 to −0.12), relatives (RD=−0.08, 95% CI=−0.11 to −0.06), neighbours/friends/colleagues (RD=−0.11, 95% CI=−0.13 to −0.09) and receiving home care (RD=−0.20, 95% CI=−0.22 to −0.18) showed significantly lower RD for depression, with the differences being slightly higher among women compared with men. For instance, the RD in depression for receiving help from

own children was −0.10 (95% CI=−0.12 to −0.08) among men compared with −0.12 (95% CI=−0.14 to −0.11) among women.

Receiving help from own children (RD=−0.10, 95% CI=−0.12 to −0.09), relatives (RD=−0.05, 95% CI=−0.07 to −0.03), neighbours/friends/colleagues (RD=−0.08, 95% CI=−0.10 to −0.06), and receiving home care (RD=−0.15, 95% CI=−0.17 to −0.13) showed significantly lower RD for nervousness. In the gender-stratified analysis, receiving help from own children showed a relatively higher RD among women, whereas receiving help from neighbours/friends/colleagues showed a relatively higher RD among men.

Receiving help from own children (RD=−0.13, 95% CI=−0.14 to −0.11) relatives (RD=−0.08, 95% CI=−0.10 to −0.05) neighbours/friends/colleagues(RD=−0.12, 95% CI=−0.13 to −0.10) and receiving home care (RD=−0.12, 95% CI=−0.14 to −0.10) showed significantly lower RD for loneliness. The RD was relatively higher among men who reported receiving help from own children, relatives and neighbours/friends/colleagues.

Receiving help from own children (RD=−0.10, 95% CI=−0.11 to −0.09), relatives (RD=−0.05, 95% CI=−0.08 to −0.03), neighbours/friends/colleagues (−0.07, 95% CI=−0.09 to −0.05) and receiving home care (RD=−0.12, 95% CI=−0.14 to −0.10) showed significantly lower RD for sleep trouble. In the gender-stratified analysis, receiving help from own children and from neighbours/friends/colleagues showed a relatively higher RD among men.

## DISCUSSION
Given the rising burden of mental illness during the COVID-19 pandemic, it is important to explore the potential role of the social determinants that can affect mental health, especially among older adults who share a greater vulnerability to both COVID-19 and mental health conditions. The present study aimed to assess the relationship between social support and indicators of psychosocial well-being among older men and women in 27 countries in Europe and Israel using data from the Survey of Health Ageing, and Retirement. Our findings suggest that about one-third of the participants reported depression, nervousness, loneliness and sleep trouble, with the percentage of all four conditions being relatively higher among women. Overall, about a quarter of the participants reported at least one, and about 8% reported having all of the four conditions. However, the percentage of reporting multiple conditions was higher among women than among men, indicating a greater burden of psychological issues among women. Previous studies have reported a higher burden of mental health issues among women and attempted to explain the gender differences in mental disorders through various biological (pregnancy, hormonal) and psychosocial factors (self-esteem, social empowerment).[23–25] Apart from the gender differences, we observed remarkable intercountry differences in the corresponding percentages as well. For instance,

## Depression(%)

| Country | % |
|---|---|
| Portugal | 46.44 |
| Bulgaria | 42.5 |
| Hungary | 42.33 |
| Poland | 39.99 |
| Malta | 38.69 |
| Latvia | 38.59 |
| Lithuania | 38.28 |
| Italy | 36.76 |
| France | 35.89 |
| Estonia | 35.07 |
| Croatia | 34.23 |
| Romania | 33.3 |
| Luxembourg | 31.65 |
| Slovakia | 31.28 |
| Spain | 31.24 |
| Israel | 29.59 |
| Germany | 29.51 |
| Greece | 29.4 |
| Belgium | 28.64 |
| Austria | 28.31 |
| Czech Republic | 26.56 |
| Cyprus | 25.1 |
| Slovenia | 23.65 |
| Finland | 23.53 |
| Switzerland | 23.18 |
| Netherlands | 19.8 |
| Sweden | 19.15 |
| Denmark | 16.7 |

## Nervousness(%)

| Country | % |
|---|---|
| Portugal | 53.37 |
| Bulgaria | 50.83 |
| Malta | 49.64 |
| Latvia | 46.07 |
| Lithuania | 41.25 |
| Israel | 40.11 |
| Hungary | 39.58 |
| Poland | 38.71 |
| Finland | 38.33 |
| France | 38.16 |
| Italy | 37.99 |
| Greece | 35.5 |
| Croatia | 34.87 |
| Luxembourg | 33.97 |
| Belgium | 33.65 |
| Spain | 32.57 |
| Romania | 32.24 |
| Estonia | 31.32 |
| Slovenia | 29.41 |
| Cyprus | 28.6 |
| Slovakia | 27.84 |
| Czech Republic | 26.77 |
| Austria | 25.03 |
| Germany | 24.67 |
| Switzerland | 21.53 |
| Sweden | 19.95 |
| Denmark | 18.68 |
| Netherlands | 14.53 |

## Loneliness(%)

| Country | % |
|---|---|
| Greece | 52.91 |
| Latvia | 51.04 |
| Bulgaria | 51.04 |
| Italy | 47.01 |
| Slovakia | 45.47 |
| Croatia | 43.48 |
| Romania | 37.08 |
| Hungary | 36.97 |
| Lithuania | 36.8 |
| Cyprus | 36.72 |
| Israel | 36.35 |
| France | 33.06 |
| Estonia | 32.49 |
| Portugal | 31.81 |
| Malta | 30.83 |
| Czech Republic | 29.99 |
| Poland | 29.56 |
| Belgium | 29.13 |
| Slovenia | 26.34 |
| Spain | 25.8 |
| Luxembourg | 24.61 |
| Sweden | 23.85 |
| Germany | 21.66 |
| Finland | 21.18 |
| Switzerland | 21.09 |
| Austria | 19.03 |
| Netherlands | 17.06 |
| Denmark | 11.49 |

## Sleep Trouble(%)

| Country | % |
|---|---|
| Estonia | 49.13 |
| Lithuania | 47.8 |
| Latvia | 47.67 |
| Israel | 46.51 |
| Bulgaria | 45.62 |
| Poland | 43.54 |
| Portugal | 42.77 |
| Romania | 36.24 |
| Slovakia | 36.16 |
| France | 34.98 |
| Croatia | 34 |
| Hungary | 33.29 |
| Germany | 32.99 |
| Slovenia | 31.83 |
| Finland | 31.77 |
| Luxembourg | 31.61 |
| Spain | 30.28 |
| Italy | 29.98 |
| Czech Republic | 29.88 |
| Belgium | 28.88 |
| Austria | 28.43 |
| Malta | 24.45 |
| Switzerland | 23.66 |
| Sweden | 23.5 |
| Greece | 21.73 |
| Denmark | 21.12 |
| Cyprus | 20.33 |
| Netherlands | 15.2 |

**Figure 1** Percentage of participants who reported having depression, nervousness, loneliness and sleep trouble by country. As shown in figure 1, the percentage of self-reported depression was over 40% in Portugal (46.44%), Bulgaria (42.5%) and Hungary (42.33%). Portugal (53.37%), Bulgaria (50.83%) also had the highest percentages of self-reported nervousness. The percentage of loneliness was highest in Greece (52.91%) and that of sleep trouble was highest in Estonia (49.13%).

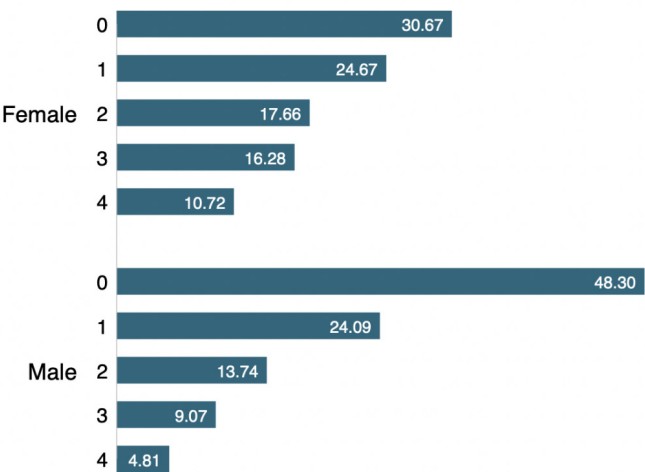

**Figure 2** Comparison of the total number of conditions reported by male and female participants. Figure 2 shows that about a quarter of the men and women reported at least one of the four conditions. However, the percentage of reporting more than one condition was higher among women than men. For example, 10.72% of the women reported having all four conditions compared with 4.81% among men.

the percentage of depression ranged from 46.44% in Portugal to 16.7% in Denmark, nervousness from 53.37% in Portugal to 14.53% in the Netherlands, loneliness from 52.91% in Greece to 11.49% in Denmark, and sleep trouble from 49.13% in Estonia to 15.2% in the Netherlands. The country-level disparities might be rooted in the health policies related to the pandemic and health and social conditions that are relevant to mental health in individual countries that should be investigated through more in-depth studies.

Regarding the association between social support in obtaining the necessities during the pandemic and the outcome measures, receiving help from own children, relatives, neighbours/friends/colleagues and home care showed significantly lower risks of depression, nervousness, loneliness and sleep trouble compared with those did not receive any help. Several previous studies have reported the positive association between social support and psychosocial well-being through a social capital lens. According to a cohort study conducted in Quebec, Canada, adolescents who felt they had more social support 1 year later reported having fewer mental health

**Table 2** Association between social support and self-reported depression, nervousness, loneliness and sleep trouble

| | Full sample | Men | Women |
|---|---|---|---|
| Depression | | | |
| Received help from own children (yes) | ref | ref | ref |
| No | −0.13*** (−0.14, −0.12) | −0.10*** (−0.12, −0.08) | −0.12*** (−0.14, −0.11) |
| Received help from relatives (yes) | ref | ref | ref |
| No | −0.08*** (−0.11, −0.06) | −0.07*** (−0.11, −0.03) | −0.08*** (−0.11, −0.05) |
| Received help from neighbours/friends/colleagues (yes) | ref | ref | ref |
| No | −0.11*** (−0.13, −0.09) | −0.09*** (−0.12, −0.06) | −0.11*** (−0.13, −0.08) |
| Received home care (yes) | ref | ref | ref |
| No | −0.20*** (−0.22, −0.18) | −0.19*** (−0.22, −0.15) | −0.19*** (−0.21, −0.16) |
| Nervousness | | | |
| Received help from own children (yes) | ref | ref | ref |
| No | −0.10*** (−0.12, −0.09) | −0.10*** (−0.11, −0.08) | −0.09*** (−0.11, −0.08) |
| Received help from relatives (yes) | ref | ref | ref |
| No | −0.05*** (−0.07, −0.03) | −0.04* (−0.08, −0.01) | −0.05*** (−0.08, −0.02) |
| Received help from neighbours/friends/colleagues (yes) | ref | ref | ref |
| No | −0.08*** (−0.10, −0.06) | −0.06*** (−0.09, −0.02) | −0.09*** (−0.11, −0.06) |
| Received home care (yes) | ref | ref | ref |
| No | −0.15*** (−0.17, −0.13) | −0.15*** (−0.18, −0.11) | −0.15*** (−0.17, −0.12) |
| Loneliness | | | |
| Received help from own children (yes) | ref | ref | ref |
| No | −0.13*** (−0.14, −0.11) | −0.09*** (−0.11, −0.08) | −0.12*** (−0.13, −0.10) |
| Received help from relatives (yes) | ref | ref | ref |
| No | −0.08*** (−0.10, −0.05) | −0.07*** (−0.11, −0.03) | −0.08*** (−0.10, −0.05) |
| Received help from neighbours/friends/colleagues (yes) | ref | ref | ref |
| No | −0.12*** (−0.13, −0.10) | −0.07*** (−0.10, −0.04) | −0.12*** (−0.15, −0.10) |
| Received home care (yes) | ref | ref | ref |
| No | −0.12*** (−0.14, −0.10) | −0.13*** (−0.16, −0.10) | −0.10*** (−0.12, −0.07) |
| Sleep trouble | | | |
| Received help from own children (yes) | ref | ref | ref |
| No | −0.10*** (−0.11, −0.09) | −0.07*** (−0.09, −0.05) | −0.09*** (−0.10, −0.08) |
| Received help from relatives (yes) | ref | ref | ref |
| No | −0.05*** (−0.08, −0.03) | −0.06** (−0.10, −0.02) | −0.04** (−0.07, −0.01) |
| Received help from neighbours/friends/colleagues (yes) | ref | ref | ref |
| No | −0.07*** (−0.09, −0.05) | −0.06*** (−0.09, −0.03) | −0.07*** (−0.09, −0.04) |
| Received home care (yes) | ref | ref | ref |
| No | −0.12*** (−0.14, −0.10) | −0.13*** (−0.16, −0.10) | −0.10*** (−0.12, −0.07) |

N.B.: Cells represent risk differences with 95% CIs. All models are adjusted for age, sex, employment status, subjective health status and country of residence.
*p<0.05, **p<0.01, ***p<0.001.

issues, concluding that perceived social support, especially in those who have mental health issues in youth, may shield against mental health issues as people enter adulthood.[19] A meta-analysis involving 64 studies reported a mean effect size of 0.356, indicating a moderate effect size of social support on mental health.[16] Another study conducted in Lebanon using structured questionnaires for depression (Patient Health Questionnaire), loneliness (University of California-Los Angeles Loneliness Scale) and social support (Multidimensional Scale of Perceived Social Support) found that the rates of depression and loneliness were significantly higher among participants who experienced self-isolation compared with those who did not.[26] The same study reported that the risk

of depressive symptoms and poor sleep quality were, respectively, 63% and 52% lower among individuals who reported higher levels of social support compared with those with low perceived social support.[26] In the context of the COVID-19 pandemic, receiving social support was reported to be inversely associated with mental health problems in several countries including China,[27] Spain,[28] Poland and Ukraine.[29]

The findings of the current study are consistent with previous ones regarding the positive role of social support on psychological well-being. Social support can take many forms, from family and friends to community groups and online forums, and serves the purpose of providing a sense of connection and belonging and improving coping skills, which can be vital for people who are feeling isolated. Lack of social support can, therefore, exacerbate the effects of social isolation and lead to further mental health problems. Arguably, the social stressors associated with the pandemic can lead to short-term and long-term psychological issues such as anxiety and depression, which in turn can contribute to feelings of loneliness and isolation. With the increasing demands of modern life, individuals are becoming more aware of the protective effects of relationships on health and overall quality of life as it can act as a buffer against social stressors and promote positive mental well-being. Mental health experts have also alarmed about the dangers of social isolation since the start of the COVID-19 pandemic. Policy-makers and social researchers should, therefore, aim to develop innovative strategies for promoting this invaluable resource to build more resilient communities that can help mitigate the long-term mental health impacts of the pandemic.

As far as we are concerned, this is the first study to focus on the association between social support and psychosocial well-being among the older adults during the COVID-19 pandemic in EU countries. The sample size was large and the data were obtained from a reputed population survey—the largest pan-European study that provides internationally comparable data on a large number of health and social indicators. The present study included participants aged 65 years and older, and the analysis was conducted using several dimensions of psychosocial well-being and social support. An important aspect of the study was the gender-stratified analysis of the association between psychosocial well-being and social support, which allowed a better understanding of the differences in the way men and women may respond to certain types of social support. Previous studies have underscored that gender-stratified analysis allows better insights into how biological, physiological, behavioural, environmental and social factors interact differently in people of different genders.[30–32] As such, gender-stratified analysis is becoming increasingly important as researchers strive to improve the understanding of the unique healthcare needs of men and women. The importance of social support for psychological well-being is a well-established finding in previous literature. However, in the present study, it was not possible to directly compare this phenomenon with the prepandemic time due to the unavailability of similar data. This study has several important limitations to consider as well. First, the data used in this analysis are cross-sectional, and therefore, no causal relationship can be established between the outcome and explanatory variables. Since the analysis was based on pre-existing data, the authors had no control over the selection and measurement of the study variables. As a result, not all dimensions of social support that are relevant to psychosocial well-being were possible to include in the analysis. Second, there was no specific information on what type of support was received by the participants from their contacts (eg, financial, activities of daily living, emotional, healthcare visits) which could have produced a more detailed picture of the association between social support and psychosocial well-being. The analysis cannot clarify the nature or quality of the relationships as well. Someone receiving all the necessary support during the pandemic may experience higher health benefits compared with those receiving the support occasionally. The measures of psychosocial well-being are also self-reported and therefore remain subject to reporting bias. Future studies should focus on addressing these limitations and use more objective indicators of social support and mental health. Regardless of these limitations, these findings make an important contribution to the literature on the positive role of social capital on psychosocial well-being. As the pandemic is starting to exert long-term social crises, which in turn are affecting people's health status, policy-makers should attempt to make socially tailored decisions to minimise the effect of social isolation and provide support, especially for the most vulnerable communities.

## CONCLUSION

According to the findings, about one-third of the participants reported having nervousness, depression, loneliness and sleep trouble. A quarter of them reported at least one, and about 8% reported having all of the four conditions, with the percentages being relatively higher among women than among men. A considerable intercountry difference was found in the corresponding percentages as well. Receiving help from own children, relatives, neighbours/friends/colleagues and home care showed significantly lower risks of depression, nervousness, loneliness and sleep trouble compared with those who did not receive any help. Interestingly, the RDs for all four indicators varied between men and women concerning the sources of social support such that the benefits are generally more pronounced among men than women.

**Contributors** Conceptualisation, JL; methodology, JX, GB, ST and SG; software, JX, GB, ST and SG; formal analysis, JX, GB, ST and SG; data curation, GB; writing—original draft preparation, JL, JX, GB, ST and SG; writing—review and editing JL, JX, GB, ST and SG. All authors have read and agreed to the published version of the manuscript. SG accepts full responsibility for the work, GB had access to the data.

**Funding** This work was funded by grant 2020YFC2006000 from the National Key R&D Programme of China, grant 72004073 from the National Natural Science

Foundation of China, and grant 20YJC630134 from the Chinese Ministry of Education of Humanities and Social Science project.

**Disclaimer** The funders had no role in the design or decision to publish the manuscript.

**Competing interests** None declared.

**Patient and public involvement** Patients and/or the public were not involved in the design, or conduct, or reporting, or dissemination plans of this research.

**Patient consent for publication** Not applicable.

**Provenance and peer review** Not commissioned; externally peer reviewed.

**Data availability statement** Data are available in a public, open access repository. Statement data are available upon reasonable reasonable request. Survey of Health, Ageing and Retirement in Europe. DOI: 10.6103/SHARE.w9ca.800. 2022-02-10.

**ORCID iDs**
Ji Lu http://orcid.org/0000-0002-2506-7223
Juyang Xiong http://orcid.org/0000-0003-0073-1826
Shangfeng Tang http://orcid.org/0000-0001-8178-2486
Ghose Bishwajit http://orcid.org/0000-0003-4461-3821

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
