## [Reviewer comments · BMJ Open]

ARTICLE DETAILS

TITLE (PROVISIONAL)	Social support and psychosocial well-being among older adults in Europe during the COVID-19 pandemic: a cross-sectional study
AUTHORS	Lu, Ji; Xiong, Juyang; Tang, Shangfeng; Bishwajit, Ghose; Guo, Shuyan

VERSION 1 – REVIEW

REVIEWER	Joseph, Letha Mullamkuzhy Duke University School of Nursing
REVIEW RETURNED	08-Feb-2023

GENERAL COMMENTS	The authors did an excellent job of highlighting the role of social support in older adults' psychosocial well-being. The topic is relevant as social isolation is common among older adults, irrespective of the pandemic. I am honored to be a reviewer. Few minor suggestions as follows: Title- In the title, "Social Support and -----Elderly men and women-----Pandemic" My thoughts- Do we need to specify men and women or can we say "Aging Population" or "Elderly Population"? Page 6, lines 26-31 I feel we should not assume that older adults generally have higher dependence and need for physical and emotional care. Did the authors mean higher dependence on physical and emotional care? On page 6, line 43 Authors mention, "studies have" And cited one reference. You may need more than one. Page 6, line 45 The cited reference (13) does not support the argument. Also, see the comment above. COVID-19 negatively impacted mental health. We assumed older adults to be the highly affected age group. However, some studies reported older adults' resilience over other age groups. So, I will carefully highlight the pandemic's impact on older adults' mental health. Page 7 The cited reference (14) does not adequately support the statement. Reference 14 is a position/feature article on older adults in Canada. Unsure that this reference is sufficient to support this strong argument.
---

REVIEWER	Koivunen, Kaisa University of Jyväskylä
REVIEW RETURNED	20-Apr-2023

GENERAL COMMENTS	This is an interesting study aiming to explore the association between social support and psychosocial well-being among older men and women in Europe during the COVID-19 pandemic. There
---

are some limitations and points that need further clarification, which are summarized below:

1. The term “elderly” is used in the article. Although it is commonly used, it has an ageist connotation and it is not recommended to be used in many journals (see e.g., Use of the Term “Elderly” : Journal of Geriatric Physical Therapy (lww.com), DOI: 10.1519/JPT.0b013e31823ab7ec). Therefore, I suggest replacing the term with more appropriate expressions, such as, “older adults/persons”.

2. Page 6, lines 42-53. The authors state that studies have revealed that older people are more likely to suffer from mental health issues compared to their younger counterparts, but only one study has been referred to. However, literature also shows the opposite results suggesting that older age groups have done better psychologically in the face of COVID-19 restrictions. For example, a review article based on 135 articles summarized that older adults experienced negative outcomes, but to a lesser extent than their younger counterparts (Lebrasseur A. 2021). It has been hypothesized that this is because younger adults may not have as developed coping mechanisms as older individuals and younger adults are often more active, and therefore, the restrictions affected more their daily lives. I therefore suggest that also this study refers to the literature on this issue more broadly than on the basis of a single research finding.

Reference:

Lebrasseur A, Fortin-Bédard N, Lettre J, Raymond E, Bussi eres EL, Lapierre N, Faieta J, Vincent C, Duchesne L, Ouellet MC, Gagnon E, Tourigny A, Lamontagne M E, Routhier F. Impact of the COVID-19 Pandemic on Older Adults: Rapid Review. JMIR Aging. 2021 Apr 12;4(2):e26474. doi: 10.2196/26474. PMID: 33720839; PMCID: PMC8043147.

3. Statistical analyses: Was the risk exposure getting social support or the lack of it in the GLM analyses? This is not entirely clear from the text.

4. My main concern is related to the lack of comparison of the results with the time before the pandemic. I think it would be important to discuss whether e.g., the prevalence of psychological conditions differ from the study results concerning time before the COVID-19 restrictions and whether the social support had more pronounced role during pandemic in maintaining psychological wellbeing. The importance of social support for psychological well-being is already known from previous literature, and therefore, I think it would be important to compare how this phenomenon manifested itself and was different during the pandemic and social isolation. And maybe also mentioned in the limitations that the study design did not allow to study this.

5. It would be important to describe somehow, what kind of social isolation measures were in place in different Europe countries at the time of the data collection and how do you think they are reflected in the results?

6. It would also be useful to describe the participation rates by country and whether there is information on non-participants. Although SHARE is a well-known study, it would be useful to describe here the recruitment process of the participants.

	7. If this was the first study on studying the association between social support and mental health in EU countries, would it be possible to compare the results with studies on the same topic in other parts of the world, e.g. UK and China, that were conducted during COVID-19 among older adults? I did find these with a quick literature search and in many studies, social support seems to be included in coping strategies that have been studied. Page 23, lines 16-44: Also , the discussion should focus on the target group, older adults, among whom the role of social support for mental health has certainly been studied. Now, the authors have referred to a study conducted with adolescents and other studies, without indicating which age groups have been targeted in these articles.
--	---

REVIEWER	Thuku, Pauline Karatina University
REVIEW RETURNED	08-May-2023

GENERAL COMMENTS	The title is well stated, and the study appropriate at a time when COVID-19 was a major health crises. The background and problem of study are well developed, methodology well described and analysis well done. However, a few observations have been made and suggestions for improvement recommended, as follows.  1. Page 5, last sentence ‘The fear of contracting the virus has forced people stay at home for a long period of time, which leads more severe among elderly individuals who share a heightened risk of severe illness from the virus’ needs to be rephrased to give meaningful communication. 2. The common practice for in-text citations is that citations at the end of a sentence should be inserted before the full stop and not after. Confirm the recommended format based on style used, for example on page 5, third line of introduction, should it be ‘livelihoods worldwide. [1–3]’ or ‘livelihoods worldwide[1–3]. –note the position of full stop! This has been observed throughout the document. 3. Page 6, first sentence ‘The measures to controll COVID-19 pandemic by physical distancing have caused disruptions a healthy and active lifestyle, and’ either has a missing word or needs rephrasing to be meaningful. The word control in the same sentence is mis-spelt and needs correction. 4. On page 6, last sentence, ‘elderly individuals are less opportunities to talk...’, the word ‘are’ should be replaced with ‘have’ 5. Page 9, first line, last word ‘iby’ should be corrected 6. Pages 12 and 13, readers need to be directed to where they can find Figures 1 and 2.
--

VERSION 1 – AUTHOR RESPONSE

Reviewer: 1

Dr. Letha Mullamkuzhy Joseph, Duke University School of Nursing, Durham VA Health Care System

Comments to the Author:

The authors did an excellent job of highlighting the role of social support in older adults' psychosocial well-being. The topic is relevant as social isolation is common among older adults, irrespective of the pandemic. I am honored to be a reviewer.

Few minor suggestions as follows:

Title- In the title, "Social Support and -----Elderly men and women-----Pandemic"

My thoughts- Do we need to specify men and women or can we say "Aging Population" or "Elderly Population"?

Response: Thanks indeed for your time to review our paper!

We changed the term to "older adults" in the title.

Page 6, lines 26-31 I feel we should not assume that older adults generally have higher dependence and need for physical and emotional care. Did the authors mean higher dependence on physical and emotional care?

Response: Thanks so much for the correction. The statement was updated as: higher dependence on physical and emotional care.

On page 6, line 43 Authors mention, "studies have" And cited one reference. You may need more than one. Page 6, line 45 The cited reference (13) does not support the argument. Also, see the comment above.

Response: Thanks for the recommendation. The sentence was deemed incongruent and was removed.

COVID-19 negatively impacted mental health. We assumed older adults to be the highly affected age group. However, some studies reported older adults' resilience over other age groups. So, I will carefully highlight the pandemic's impact on older adults' mental health.

Page 7 The cited reference (14) does not adequately support the statement. Reference 14 is a position/feature article on older adults in Canada. Unsure that this reference is sufficient to support this strong argument.

Response: This statement was also removed due to lack of supporting evidence.

Reviewer: 2

Dr. Kaisa Koivunen, University of Jyväskylä

Comments to the Author:

This is an interesting study aiming to explore the association between social support and psychosocial well-being among older men and women in Europe during the COVID-19 pandemic. There are some limitations and points that need further clarification, which are summarized below:

1. The term "elderly" is used in the article. Although it is commonly used, it has an ageist connotation and it is not recommended to be used in many journals (see e.g., Use of the Term "Elderly" : Journal of Geriatric Physical Therapy ([lww.com](http://www.com)), DOI: 10.1519/JPT.0b013e31823ab7ec). Therefore, I suggest replacing the term with more appropriate expressions, such as, "older adults/persons".

Response: Thanks indeed for the correction. The term was replaced with 'older adults' in the text.

2. Page 6, lines 42-53. The authors state that studies have revealed that older people are more likely to suffer from mental health issues compared to their younger counterparts, but only one study has been referred to. However, literature also shows the opposite results suggesting that older age groups have done better psychologically in the face of COVID-19 restrictions. For example, a review article

based on 135 articles summarized that older adults experienced negative outcomes, but to a lesser extent than their younger counterparts (Lebrasseur A. 2021). It has been hypothesized that this is because younger adults may not have as developed coping mechanisms as older individuals and younger adults are often more active, and therefore, the restrictions affected more their daily lives. I therefore suggest that also this study refers to the literature on this issue more broadly than on the basis of a single research finding.

Reference:

Lebrasseur A, Fortin-Bédard N, Lettre J, Raymond E, Bussi eres EL, Lapierre N, Faieta J, Vincent C, Duchesne L, Ouellet MC, Gagnon E, Tourigny A, Lamontagne M E, Routhier F. Impact of the COVID-19 Pandemic on Older Adults: Rapid Review. *JMIR Aging*. 2021 Apr 12;4(2):e26474. doi: 10.2196/26474. PMID: 33720839; PMCID: PMC8043147.

Response: Thanks so much for sharing the reference! The suggestions are very insightful. We updated text accordingly and removed the conflicting statements.

3. Statistical analyses: Was the risk exposure getting social support or the lack of it in the GLM analyses? This is not entirely clear from the text.

Response: We agree that this not obvious from risk differences. The reference category was 'having the support.

4. My main concern is related to the lack of comparison of the results with the time before the pandemic. I think it would be important to discuss whether e.g., the prevalence of psychological conditions differ from the study results concerning time before the COVID-19 restrictions and whether the social support had more pronounced role during pandemic in maintaining psychological wellbeing. The importance of social support for psychological well-being is already known from previous literature, and therefore, I think it would be important to compare how this phenomenon manifested itself and was different during the pandemic and social isolation. And maybe also mentioned in the limitations that the study design did not allow to study this.

Response: Thanks for this comment. If the previous surveys collected data on these variables, it could have been immensely helpful in comparing the data with the pandemic period. This limitation was acknowledged in the discussion to provide transparency and clarify the constraints of the study design.

5. It would be important to describe somehow, what kind of social isolation measures were in place in different Europe countries at the time of the data collection and how do you think they are reflected in the results?

Response: This is an interesting idea. We have thought through it, however not sure how to incorporate the country-level situations given the number of countries included in the analysis. News reports say that all the countries imposed strict lockdown measures during the pandemic period which has resulted in numerous mental health and social issues. Our first guess was that people who remained more socially active and received assistance enjoyed better mental health status.

6. It would also be useful to describe the participation rates by country and whether there is information on non-participants. Although SHARE is a well-known study, it would be useful to describe here the recruitment process of the participants.

Response: Thanks for the suggestion. Sampling method was mentioned in the methods (data source) section.

7. If this was the first study on studying the association between social support and mental health in EU countries, would it be possible to compare the results with studies on the same topic in other parts of the world, e.g. UK and China, that were conducted during COVID-19 among older adults? I did find these with a quick literature search and in many studies, social support seems to be included in coping strategies that have been studied.

Response: As far as we are concerned, this is the first original study studying the association between social support and mental health in the EU countries for this age groups. We have cited a few more studies in the discussion section for different age groups. Thanks for the suggestion.

Page 23, lines 16-44: Also , the discussion should focus on the target group, older adults, among whom the role of social support for mental health has certainly been studied. Now, the authors have referred to a study conducted with adolescents and other studies, without indicating which age groups have been targeted in these articles.

Response: Thanks for this comment. We share the same concern with you. The lack of comparable studies based on similar population groups was a major issue we faced in composing the discussion section. We nonetheless cited those studies to provide a broad understanding of the association, regardless of the age or setting, as none of the studies have actually focused on older adults.

Reviewer: 3

Dr. Pauline Thuku, Karatina University

Comments to the Author:

The title is well stated, and the study appropriate at a time when COVID-19 was a major health crises. The background and problem of study are well developed, methodology well described and analysis well done. However, a few observations have been made and suggestions for improvement recommended, as follows.

1. Page 5, last sentence 'The fear of contracting the virus has forced people stay at home for a long period of time, which leads more severe among elderly individuals who share a heightened risk of severe illness from the virus' needs to be rephrased to give meaningful communication.

Response: Thanks for your comment. The sentence was rephrased for clarity: The fear of contracting the virus has compelled people to stay at home for an extended period of time, particularly affecting elderly individuals who face an elevated risk of severe illness from the virus.

2. The common practice for in-text citations is that citations at the end of a sentence should be inserted before the full stop and not after. Confirm the recommended format based on style used, for example on page 5, third line of introduction, should it be 'livelihoods worldwide. [1–3]' or 'livelihoods worldwide[1–3]. –note the position of full stop! This has been observed throughout the document.

Response: Thanks for the suggestion. Reference style was now updated to match the requirements of BMJ Open.

3. Page 6, first sentence 'The measures to controll COVID-19 pandemic by physical distancing have caused disruptions a healthy and active lifestyle, and' either has a missing word or needs rephrasing to be meaningful. The word control in the same sentence is mis-spelt and needs correction.

Response: The sentence was reworded: The measures to control the COVID-19 pandemic, such as physical distancing, have caused disruptions to a healthy and active lifestyle, as well as the opportunities to maintain social connections and seek help in times of need.

4. On page 6, last sentence, 'elderly individuals are less opportunities to talk...', the word 'are' should be replaced with 'have'

Response: The error was corrected. Thank you!

5. Page 9, first line, last word 'iby' should be corrected

Response: The typo was corrected. Thank you!

6. Pages 12 and 13, readers need to be directed to where they can find Figures 1 and 2.

Response: Thanks for the suggestion. We have uploaded the figures according to the journal guidelines.

VERSION 2 – REVIEW

REVIEWER	Koivunen, Kaisa University of Jyväskylä
REVIEW RETURNED	06-Jun-2023
GENERAL COMMENTS	The authors have addressed all the points I raised and I have no further comments.